# Hyaluronan-Induced CD44-iASPP Interaction Affects Fibroblast Migration and Survival

**DOI:** 10.3390/cancers15041082

**Published:** 2023-02-08

**Authors:** Chun-Yu Lin, Kaustuv Basu, Aino Ruusala, Inna Kozlova, Yan-Shuang Li, Spyridon S. Skandalis, Carl-Henrik Heldin, Paraskevi Heldin

**Affiliations:** 1Department of Medical Biochemistry and Microbiology, Uppsala University, SE-751 23 Uppsala, Sweden; 2Division of Infectious Diseases, Department of Internal Medicine, Kaohsiung Medical University Hospital, School of Medicine, College of Medicine, Center for Tropical Medicine and Infectious Diseases Research, Kaohsiung Medical University, Kaohsiung 807, Taiwan; 3Department of Mechanical Engineering, McGill University, Montréal, QC H3A 0C3, Canada; 4Department of Breast Surgery, Beijing Chaoyang Hospital, Capital Medical University, Beijing 100020, China; 5Laboratory of Biochemistry, Department of Chemistry, University of Patras, 26504 Patras, Greece

**Keywords:** CD44, p53, iASPP, hyaluronan, apoptosis, cancer, tumor microenvironment

## Abstract

**Simple Summary:**

The hyaluronan receptor CD44 plays a vital role in tumor cell growth and chemotherapy resistance. The tumor suppressor p53 binds to the CD44 promoter via a non-canonical p53 consensus sequence and suppresses CD44 expression, promoting apoptosis. Conversely, the apoptotic function of p53 is negatively regulated through its interactions with iASPP, a specific inhibitor of p53-induced apoptosis in both normal and cancer cells. CD44, p53, and iASPP are co-expressed in a wide array of human cancers; however, the precise role of CD44 in p53-mediated apoptosis is not known. We discovered that the standard isoform of CD44 physically bound to iASPP and dictated the sub-cellular localization of iASPP-p53 complexes. The iASPP-CD44 complex affected fibroblast adhesion, migration, growth, and p53-mediated apoptosis.

**Abstract:**

In the present study, we show that the inhibitor of the apoptosis-stimulating protein of p53 (iASPP) physically interacts with the hyaluronan receptor CD44 in normal and transformed cells. We noticed that the CD44 standard isoform (CD44s), but not the variant isoform (CD44v), bound to iASPP via the ankyrin-binding domain in CD44s. The formation of iASPP-CD44s complexes was promoted by hyaluronan stimulation in fibroblasts but not in epithelial cells. The cellular level of p53 affected the amount of the iASPP-CD44 complex. iASPP was required for hyaluronan-induced CD44-dependent migration and adhesion of fibroblasts. Of note, CD44 altered the sub-cellular localization of the iASPP-p53 complex; thus, ablation of CD44 promoted translocation of iASPP from the nucleus to the cytoplasm, resulting in increased formation of a cytoplasmic iASPP-p53 complex in fibroblasts. Overexpression of iASPP decreased, but CD44 increased the level of intracellular reactive oxygen species (ROS). Knock-down of CD44s, in the presence of p53, led to increased cell growth and cell density of fibroblasts by suppression of p27 and p53. Our observations suggest that the balance of iASPP-CD44 and iASPP-p53 complexes affect the survival and migration of fibroblasts.

## 1. Introduction

Hyaluronan is a polysaccharide ubiquitously found in the extracellular matrix, which regulates cell homeostasis through its interaction with hyaladherins, including the cell surface receptor CD44 [1,2]. An aberrant accumulation of hyaluronan is seen during rapid tissue remodeling, occurring during embryogenesis, inflammation, and tumor progression. Increased hyaluronan production is correlated to poor prognosis for cancer patients [3,4,5,6,7,8].

Hyaluronan binds to the transmembrane glycoprotein CD44, which occurs in several isoforms, i.e., the standard form (CD44s) and variant forms (CD44v) [9,10,11,12]. CD44 is ubiquitously found on the cell surface of mesenchymal, endothelial, and epithelial cells and it controls cellular behavior, such as homing of leukocytes [13], adhesion of fibroblasts to extracellular matrix [14], and cancer cell invasion and adhesion to endothelium [15,16,17,18,19]. Furthermore, CD44 interacts with members of the ERM (ezrin, radixin, and moesin) family [20], which have important functions in cytoskeletal organization and regulation of apoptosis [21]. In particular, the interaction between CD44 and ezrin and subsequent cytoskeletal rearrangements promotes Fas-mediated apoptosis [22,23].

Previous research has highlighted the critical importance of hyaluronan-activated CD44 in modulating the response of fibroblasts to platelet-derived growth factor (PDGF)–BB-induced cell migration, most likely by the recruitment of an active protein tyrosine phosphatase to the PDGF β-receptor [24]. As part of our effort to identify proteins that mediate the intracellular signaling of CD44, we used a proteomic approach to identify molecules binding to the cytoplasmic part of CD44 [25]. The analysis resulted in the identification of the IQ motif containing GTPase protein (IQGAP1), which is a key regulator of cell–cell and cell–matrix adhesion [26], and the inhibitor of apoptosis-stimulating protein p53 (iASPP), which inhibits p53-mediated cell death [27,28]. iASPP inhibits the transcriptional activity of p53 on the promoters of Bax and PIG3 [27]. Originally, it was described as a RelA-associated inhibitor (RAI), which inhibits the transcriptional activity of the NF-κB subunit p65 (RelA) [29,30]. iASPP belongs to the ASPP family of proteins, which also consists of ASPP1 and ASPP2, which promote p53-induced apoptosis and are found both in the nucleus and cytoplasm. iASPP is overexpressed in a variety of cancers and accumulates in the nucleus of prostate cancer cells [31,32]. p53 is a hub of signaling pathways and accumulates in the nucleus where it transcriptionally regulates apoptosis and the cell cycle; however, under cell-death conditions, p53 moves to the cytoplasm and inhibits autophagy [33]. Interestingly, induction of p53 leads to CD44 repression under basal physiologic conditions and during cell stress [34].

In the present study, we have demonstrated that CD44s forms a complex with iASPP and thereby affects the interaction between p53 and iASPP, that iASPP is needed for CD44-induced cell migration, and that CD44s suppresses cell growth.

## 2. Materials and Methods

### 2.1. DNA Constructs

A cDNA for CD44 standard form (a generous gift by Professor Ivan Stamenkovik, Lausanne University Hospital, Switzerland) was cloned into the pMSCV hygro vector (Clonetch Laboratories, Inc., Mountain View, CA, USA) with BglII and XhoI. Point mutations, S291A, S316A, or S325A, in pMSCVhyg_CD44 were inserted by a site-directed mutagenesis kit according to the manufacturer (Stratagene, La Jolla, CA, USA). The mutagenic primers used in the sense orientation were (the altered codon in bold): S291A, 5′-GCATTGCAGTCAAC**GC**TCGAAGAAGGTG-3′; S316A_,_ 5′-GACAGAAAGCCA**GC**TGGACTCAACGG-3′; S325A, 5′-GAGGCCAGCAAGG**C**TCAGGAAATGGTG-3′. For the construction of a CD44 mutant altering the putative nuclear localization signal (NLS; _291_SRRRCGQKKK_300_), PCR was performed in two steps. The first sense primer was to change _292_RRR_294_ to _292_AAA_294_ (altered codons in bold): 5′- GCATTGCAGTCAACAGT**GCAGCAGCG**TGTGGGCAGAAGAAAAGC-3′; and the second was to change _298_KKK_300_ to _298_AAA_300_: 5′- GCAGCGTGTGGGCAG**GCGGCAGCG**CTAGTGATCAACAGTGG-3′. For the construction of a CD44 deletion mutant depleted of the 40 C-terminal amino acids residues, two primers were used: 5′- GAATTAGATCTTCCCCACC-3′ (containing the BglII site of pMSCVhyg_CD44 vector) and a primer possessing TAA TGA TAG stop codons after the GGA codon, which corresponds to Gly_320_ followed by a Xho1 site (altered codons in bold): 5′-CGCCGCTCGAGCTATCATTA**TCC**GTTGAGTCCACTTG-3′. To mutate the ankyrin binding domain, ^306^NSGNGAVEDRKPSG^320^L, R^313^I and K^314^N [10,35] mutations were introduced using the forward 5′-GCT GTG GAG GAC A**T**A AA**C** CCA AGT GGA CTC AAC GG-3′ and reverse 5′-CC GTT GAG TCC ACT TGG **G**TT T**A**T GTC CTC CAC AGC-3′ primers (altered codons in bold). After mutagenesis, the PCR-amplified products were treated with DpnI (digestion of parental methylated and hemi-methylated DNA) before transformation into DH5α competent cells. Isolated DNA from colonies was screened for the desired mutations by sequencing (ABI Prism 310 genetic Analyzer, Applied Biosystems). Then, wild-type (wt) CD44 and CD44 mutants were cloned in pcDNA3 in BamHI and XhoI sites and re-sequenced.

iASPPwt-v5 construct was a generous gift from Professor Xin Lu, Ludwig Cancer Research, Oxford, England.

### 2.2. Cell Culture and Antibodies

Human embryonic kidney cells (HEK293), human foreskin fibroblasts (AG1523; Coriell Institute for Medical Research, NJ, USA), the breast cancer cell line MCF7, and the human hepatocellular liver carcinoma cell line (HepG2) were obtained from ATCC (Manassas, VA, USA). Telomerase immortalized human foreskin fibroblasts (hTERT-BJ) were kindly provided by Dr Annika Gad, Ludwig Cancer Research, Uppsala, Sweden. Human dermal fibroblasts (MTS64) were established from breast reduction surgery after approval, as described [24]. Cells were routinely cultured in Dulbecco’s modified Eagle’s medium (DMEM; Invitrogen, Walham, MA, USA), supplemented with 10% fetal bovine serum (FBS; Hyclone, GEHealth Care, Uppsala, Sweden), at 37 °C in 5% CO_2_. MCF10A human mammary epithelial cells (generous gift of Professor Aristides Moustakas, Uppsala University, Sweden) were cultured in DMEM/F12 medium (1:1; Thermo Fischer Scientific #11330-032), supplemented with 5% horse serum (Invitrogen), EGF (20 ng/mL; PeproTech EC Ltd. Nordic, Stockholm, Sweden), as well as insulin (10 µg/mL; #I-1882), hydrocortisone (0.5 µg/mL; #H-0888), and cholera toxin (100 ng/mL; #C-8052) from Sigma-Aldrich (Merck KGaA, Darmstadt, Germany).

hTERT-BJ or AG1523 cells were starved for 24 h in medium containing 0.1% FBS, before stimulation with hyaluronan (100 µg/mL; molecular mass of 1 × 10^6^; 45 min), 10% FBS, phorbol 12-myristate 13-acetate (PMA) that activates PKC (80 nM; 45 min), PDGF-BB (10 ng/mL; 10 min), Nutlin (10 nM; overnight), and TGFβ (2 ng/mL; 30 min).

The antibodies used for immunoprecipitation and immunoblotting are listed in Appendix A.

### 2.3. Transfections

HEK293 and MCF7 cells (8 × 10^6^ cells per 10 cm culture dish) were transiently transfected with 1 μg each of vectors encoding wt CD44, CD44 mutants, or wt iASPP using Lipofectamine 2000 (Invitrogen Walham, MA, USA) for 48 h in complete medium, following the instructions of the manufacturer. The ratio between cDNA and Lipofectamine was kept at 1:2. Suppression of CD44 and iASPP expression was performed by transient transfection using siRNAs specific for CD44 (20 nM for 72 h; Dharmacon, Layafette, CO, USA) or iASPP (two additions of 40 nM siRNA each time within 24 h, followed by incubation for 72 h), with a scrambled siRNA as control, using SilentFect (Bio-Rad laboratories, Solna, Sweden), according to the instructions of the manufacturer.

### 2.4. Cell Lysis, Subcellular Fractionation, Immunoblotting, and Co-Immunoprecipitation

Cells grown in complete media were washed with ice-cold phosphate buffer saline (PBS; pH 7.2). To prepare cell lysates, cells were scraped in lysis buffer (1% NP40, 0.1% SDS, 0.5% deoxycholate, 50 mM Tris, pH 8.0, and 150 mM NaCl), supplemented with protease inhibitors (0.5 µg/mL Pefabloc, 10 µM Leupeptin, 9100 KIU/mL Aprotinin) and phosphatase inhibitors (50 mM NaF, 1 mM orthovanadate), and incubated on ice for 30 min. Following centrifugation (13,000 rpm, 15 min, 4 °C), the supernatants were collected. Protein amount was measured by a BCA kit (Thermo Fisher Scientific, Gothenburg, Sweden). For nuclear and cytoplasmic fractionation, hTERT-BJ cells expressing or depleted of CD44 were grown in 100 mm Petri dishes, lysed in cytoplasmic lysis buffer, followed by centrifugation at 13,000 rpm for 20 min to separate cytoplasmic fraction (supernatant) from nuclear fraction (pellet) [36]. Samples of equal protein content were subjected to SDS-polyacrylamide gel electrophoresis (SDS-PAGE), followed by wet transfer to Hybond Extra nitrocellulose membranes (Amersham, GE Healthcare, Sweden) at 100 V for 90 min at 4 °C. Membranes were blocked in 5% bovine serum albumin (BSA) or 5% milk in Tris-buffered saline (TBS), supplemented with 1% Tween 20 (TBST), and incubated with antibodies (listed in Appendix A) at 4 °C overnight. Immunocomplexes were detected by incubation with horseradish peroxide-conjugated secondary goat anti-rabbit IgG (1:10,000 dilution; Thermo Fisher Scientific, Gothenburg, Sweden, Cat# 65-6120, RRID:AB_2533967) and goat anti-mouse IgG (1:10,000 dilution; Thermo Fisher Scientific, Gothenburg, Sweden, Cat# 65-6120, RRID:AB_228307) for 1 h at room temperature, followed by development by chemiluminescence (Millipore, MA, USA); protein bands were scanned and quantified by Bio-Rad Universal Hood II CCD camera (Bio-Rad, Hercules, CA, USA).

For immunoprecipitation, cell lysates, containing approximately 1–2 mg protein, were precleared with 30 μL of protein G-Sepharose beads (GE Healthcare; 50% slurry in PBS) end-over-end at 4 °C for 15 min and then centrifuged at 3500 rpm for 5 min. The supernatants were collected in pre-lubricated tubes and incubated with 3 μg of primary antibody or with the corresponding amount of control IgG, end-over-end at 4 °C overnight. Then, Sepharose-G beads (50 μL) were added, followed by end-over-end mixing for 1 h at 4 °C. The samples were then centrifuged at 3500 rpm for 5 min, and the supernatants were removed. Beads were washed once with high salt washing buffer (1% NP40, 50 mM Tris, pH 8, 500 mM NaCl, and 1 mM EDTA), followed by four washes with low salt washing buffer (1% NP40, 0.1% SDS, 0.5% deoxycholate, 50 mM Tris, pH 8, and 150 mM NaCl), and finally once with PBS. Thirty microliters of reducing SDS sample buffer was added to elute the captured proteins, after which the samples were heated at 95 °C for 5 min and then centrifuged at 3500 rpm for 5 min. The supernatants were subjected to SDS-PAGE and analyzed by immunoblotting.

### 2.5. Proximity Ligation Assay

The interaction between iASPP and CD44 was also investigated using in situ proximity ligation assay (PLA) (Olink Bioscience, Uppsala, Sweden), according to the manufacturer’s protocol, essentially as described before [26].

### 2.6. In Silico Analysis

In silico homology-based protein–protein interaction analysis was performed using PSOPIA, a web-based averaged one-dependence estimator (AODE). The least probability of interaction scores 0, and the maximum possibility of protein–protein interaction scores 1 [37]. We analyzed mRNA expression data of *CD44*, *iASPP* and *p53* using clinical datasets from seven types of cancers, including head and neck squamous cell carcinoma, adult soft tissue sarcoma, invasive breast carcinoma, lung squamous cell carcinoma, liver hepatocellular carcinoma, glioblastoma multiforme, and kidney renal papillary cell carcinoma. Data were retrieved from the TCGA database using cBioPortal for Cancer Genomics, an open-access, open-source resource for multidimensional cancer genomics data sets [38,39].

### 2.7. Cell Proliferation Assay

In order to investigate the effect of silencing CD44 on cell proliferation, 100,000 AG1523 human fibroblasts, treated with 20 nM of CD44 siRNA or control scramble siRNA, were seeded into each well of a 6-well plate and cultured in medium containing 10% FBS. The suppression of CD44 was initiated at the day of plating in culture medium supplemented with 2% FBS, and an additional amount of CD44 siRNA (10 nM) was added after 24 h. Culture media were refreshed every other day with scramble or CD44 siRNAs (10 nM). Cell amounts were quantified 3 days and 6 days after initial seeding. Cell morphologies were observed with a phase contrast Zeiss Axiovision 40 microscope (Carl Zeiss AB, Stockholm, Sweden), and the cell number was counted with a Luna cell counter.

Cell proliferation was also determined by the incorporation of ^3^H-thymidine into DNA, as measured by a liquid scintillation counter.

### 2.8. Immunofluorescence Staining

hTERT-BJ fibroblasts were fixed in 3% paraformaldehyde for 20 min at room temperature. Thereafter, cells were permeabilized with 0.25% Triton X-100 for 10 min, washed in PBS, blocked by 3% BSA and 5% goat serum in PBS for 1 h, incubated with primary antibodies for 90 min at room temperature, washed four times in PBS, and incubated with secondary antibodies for 1 h. The following primary and secondary antibodies were used at the indicated dilutions and concentration: anti-p53 (1:200; mouse IgG DO-1), anti-CD44 (4 μg/mL; Hermes1), iASPP antiserum (1:200), Alexa Fluor 488 goat anti-mouse (1:1000), and Alexa Fluor 594 goat anti-rabbit (1:1000). Slides were mounted with ProLon gold antifade reagent (Invitrogen), and photographs were taken with a Zeiss Axioplan 2 immunofluorescence microscope and/or a Zeiss LSM 700 confocal module.

### 2.9. Scratch Wound Migration Assay

hTERT-BJ fibroblasts were cultured in complete medium in six-well plates for 24 h and then transfected with siRNA (40 nM) against iASPP or scramble control. After 24 h, the cells were further transfected with the same siRNAs for another 24 h. Cells were starved for 24 h, and a scratch was made manually in the monolayer with a pipette tip. Following gentle washing with PBS, cells were pretreated, or not, with a CD44 antibody (Hermes1) (20 μg/mL) for 2 h before incubation with hyaluronan (100 μg/mL) or PDGF-BB (10 ng/mL) in starvation medium for 24 h. Untreated cells were used as controls. Phase contrast images of the wounded areas were photographed at time 0 h and 24 h after stimulation. Wounded areas covered by cells were quantified using T-scratch software (CSE lab, ETH Zurich, Switzerland), and migration of the cells was determined as the part of the wounded area that had been covered by cells. Each assay was conducted in triplicate and repeated at least three times.

### 2.10. Adhesion Assay

To determine the role of CD44-iASPP complexes in cell adhesiveness, HMW hyaluronan (0.1 mg/mL) and Sulfo-NHS (0.184 mg/mL) in distilled H_2_O were used to coat a Covalink-NH 96-well microtiter plate (50 µL/well) (NUNC, Thermo Fisher Scientific, Gothenburg, Sweden, Cat. No.: #478042). Then, 50 µL/well of 0.123 mg/mL EDAC in H_2_0 was added, and plates were left in room temperature for 2 h, followed by incubation overnight at 4 °C. Each well was washed twice with 2 M NaCl for 5 min, then once with PBS, and three times with 0.05% Tween for 5 min each. The remaining fluid was aspirated, and plates were blocked with 0.5% BSA in PBS (200 µL/well) for 30 min at 37 °C, followed by a wash with PBS.

AG1523 cells depleted of iASPP or p53 by siRNA (100,000 cells/350 µL DMEM, 0.1% BSA) were collected in Eppendorf tubes and incubated with hyaluronan (100 µg/mL for 60 min), anti-CD44 Hermes1 antibodies (20 µg/mL for 30 min), PMA (100 nM; 10 min), or combinations thereof. Then, 30,000 cells/100 µL in DMEM, 0.1% BSA were seeded into each well and incubated at 37 °C for 30 min to let the cells adhere. Cells were then gently washed with PBS to remove non-adherent cells and stained with 100 µL/well 0.5% crystal violet in 20% methanol for 20 min at room temperature with gentle shaking, followed by washing four times thoroughly with distilled H_2_O to remove excess dye. Plates were then dried completely with open lid, overnight in the dark. Attached cells were retrieved by addition of 100 µL per well of 100% methanol and incubation for 20 min at room temperature with gentle shaking. Finally, the absorbance at 570 nm was measured for each well.

### 2.11. Measurement of Intracellular ROS Levels

HEK293 cells ectopically expressing iASPPwt-v5, and CD44s, alone or in combinations, were treated with 2′, 7-dichlorofluorescin diacetate (DCFH-DA; Sigma Aldrich, Gothenburg, Sweden) (5 μg/mL) for 10 min in complete darkness, to measure the production of ROS. Then, DCFH-DA was removed, cells were washed twice with ice-cold PBS, and fluorescence in DCFH-DA-loaded cells was measured at the excitation wavelength 488 nm and emission wavelength 530 nm. The fluorescence intensity from each sample was normalized with the total protein concentration (mg/mL) as quantified by BCA assay.

## 3. Results

### 3.1. The Standard Isoform of CD44 Binds to iASPP in Mesenchymal and Epithelial Cells

We previously showed that an immobilized C-terminal part of CD44 bound iASPP in a cell-free system [25]. Here, we evaluated their interaction in living cells. We first determined the expression of CD44, iASPP, and p53 by immunoblotting of cell lysates from normal and transformed cells of both mesenchymal and epithelial origin, including AG1523 fibroblasts, immortalized hTERT-BJ fibroblasts, immortalized normal mammary epithelial cells MCF10A, metastatic breast cancer cells MCF-7, HEK293 cells, non-small cell lung carcinoma cells H1299, osteosarcoma U2OS, primary dermal fibroblasts MTS64, and hepatocellular HepG2 cells (Figure 1). We observed that iASPP was abundantly expressed in H1299 and HepG2 cells, while low expression was evident in MCF-7 and MTS64. The CD44 standard (CD44s) isoform was expressed in all cell types, abundantly in H1299, MTS64, hTERT-BJ, and AG1523, and at low levels in MCF-7, HEK293T, and HepG2 cells. CD44 variant isoforms (CD44v) were observed in U2OS, H1299, and MCF-7. p53 was expressed at high levels in HEK293 and HepG2 cells and at lower level in H1299 cells. Two isoforms of p53 (P72 and R72) were expressed in MCF-7 cells and HEK293 (Figure 1A). To determine whether CD44 and iASPP interact, cell lysates from AG1523, hTERT-BJ, MCF10A, HepG2, MTS64, and H1299 cells were subjected to immunoprecipitation (IP) using an anti-iASPP antibody and IgG isotype as control, followed by immunoblotting (IB) with antibodies against CD44 and iASPP; complexes between iASPP and CD44 standard isoform (CD44s), but not with the variant form (CD44v), were seen in all cell types but at different amounts (Figure 1B,C). In addition, in overexpression experiments, immunoprecipitation of CD44 with IM7 antibodies, followed by immunoblotting with an iASPP antibody, revealed the formation of a complex (Appendix A).

### 3.2. Hyaluronan Enhanced the Amount of iASPP-CD44s Complexes in Mesenchymal Cells but Not in Epithelial Cells

Stimulation with high molecular weight hyaluronan (HMW HA; Mw 1 × 10^6^) enhanced the formation of an iASPP-CD44 complex in mesenchymal (hTERT-BJ) cells but not in epithelial (MCF10A) cells (Figure 1C), suggesting that CD44-iASPP complex formation may be induced differently in different cell types in response to external stimuli. The hyaluronan-mediated increase in the interaction between iASPP and CD44 was also confirmed in situ by proximity ligation assay (Appendix A).

### 3.3. Characterization of the Epitopes Involved in the Interaction between CD44 and iASPP

The phoshorylation status of CD44 affects its association with the cytoskeleton and, thus, cell migration; a PKC-induced dephosphorylation of S325 and concomitant phosphorylation of S291 and S316 by another kinase results in the dissociation of CD44 from activated ezrin, radixin, and moesin (ERM) proteins [10]. To investigate whether the phosphorylation status and specific regions in the cytosolic domain of CD44 affected its interaction with iASPP, we generated several CD44 mutants (Figure 2A,B) and expressed them in HEK293 cells that express only minute amounts of CD44. Cell lysates were subjected to immunoprecipitation with antibodies against iASPP, followed by immunoblotting with CD44 and iASPP antibodies (Figure 3A). An interaction between wt CD44 and iASPP was seen. Furthermore, interactions between iASPP and the CD44 phosphorylation site mutants and between the NLS mutant and the ∆320 mutant were seen, which were even stronger than that between wt CD44 and iASPP. Of note, mutation of the CD44 ankyrin binding domain (R^313^I, K^314^N) suppressed the formation of complexes between iASPP and CD44 (Figure 3B). These findings suggest that the ankyrin-binding domain ^306^NGGNGTVEDRKPSE^320^L binds iASPP and that mutations of the NLS or deletion of the C-terminal tail of CD44 promote the accessibility of the ankyrin-binding domain of CD44 for iASPP. Importantly, overexpression of p53 reduced CD44-iASPP complex formation (Figure 3A).

When we transfected the same CD44 mutants in MCF-7 breast cancer cells, we also noticed significantly less interaction between iASPP and CD44 mutated in the ankyrin-binding domain compared with CD44wt-iASPP interaction; however, in this cell line, no enhancement of binding of the NLS mutant and Δ320 mutant CD44 was seen (Figure 3C).

We performed an in silico homology-based protein–protein interaction analysis by molecular prediction server PSOPIA to delineate which part of iASPP binds to CD44. The study predicted that full-length CD44 might interact with the C-terminus of iASPP, where the ankyrin domain is located (score = 0.7394), but not with the N-terminus (score = 0.3537) (Figure 2A and Appendix A).

### 3.4. The Level of p53 in hTERT-BJ Cells Affects the Formation of the iASPP-CD44 Complex

Next, we addressed the question how the expression of p53 influenced the interaction between iASPP and CD44. Overexpression of p53 in hTERT-Bj cells strongly suppressed the complexes between CD44 and iASPP and considerably repressed the expression of CD44 (Figure 4). To further investigate a possible role of p53 in iASPP-CD44 complex formation, we depleted hTERT-BJ cells of p53 using siRNA or treated cells with Nutlin, which inhibits MDM2-p53 interaction leading to stabilization of p53. Interestingly, a stronger interaction between iASPP and CD44 was observed in cells in which p53 had been knocked-down, compared with Nutlin-treated cells expressing high levels of p53 (Figure 4).

### 3.5. CD44 Expression Modulates the Sub-Cellular Localization of iASPP-p53 Complexes

iASPP and p53 are localized both in the nucleus and in the cytoplasm; it has been hypothesized that the nuclear expression of iASPP and p53 may facilitate tumor progression [31]. We investigated how the expression of CD44 affected the cellular localization of iASPP and CD44, as well as the formation of iASPP-p53 complexes. Using immunofluorescence staining, the subcellular localization of p53 and iASPP was monitored in hTERT-BJ cells; a predominantly nuclear localization of iASPP and p53 was seen (Figure 5A). Interestingly, CD44-silenced hTERT-BJ cells displayed an enrichment of iASPP and p53 in the cytoplasm. These findings were corroborated by confocal microscopy, which demonstrated clear differences in localizations of p53 and iASPP in hTERT-BJ cells expressing CD44 compared with cells not expressing CD44; a lower number of complexes were found in the nucleus in CD44-silenced cells (Figure 5A).

To further evaluate these observations, the subcellular localization of iASPP-p53 complexes in cultures expressing CD44, or not, were also investigated by co-immunoprecipitation. The analysis revealed an approximate 2-fold increase in iASPP-p53 complexes in the cytoplasmic fraction from CD44-depleted cells compared with CD44 expressing cells, whereas the amount of nuclear iASPP-p53 complexes were decreased after knock-down of CD44 (Figure 5B).

### 3.6. Differential Expression of CD44, iASPP, and p53 in Different Tumor Types

Two ways of gene interaction, co-occurrence and mutual exclusivity, drive somatic alterations promoting tumor growth. In co-occurrence, alterations of two genes co-exist in the same tumor, whereas in mutual exclusion, one out of the two genes is altered in a tumor [40]. We studied in silico the gene–gene interactions of *CD44-iASPP* and *CD44-p53* complexes and how these interactions affect tumor survival by analyzing mRNA expression data of seven different types of cancers (Appendix A), including head and neck squamous cell carcinoma (*n* = 528), adult soft tissue sarcoma (*n* = 206), invasive breast carcinoma (*n* = 1084), lung squamous cell carcinoma (*n* = 487), liver hepatocellular carcinoma (*n* = 372), glioblastoma multiforme (*n* = 585), and kidney renal papillary cell carcinoma (*n* = 283) from the TCGA database using cBioPortal. We observed that in different types of cancers, *CD44, iASPP*, and *p53* were expressed differently. In invasive breast carcinoma, *CD44-iASPP* (*p* = 0.02) and *CD44-p53* (*p* = 0.34) were mutually exclusive, and all three genes were negatively co-expressed, whereas in liver hepatocellular carcinoma, *CD44-iASPP* (*p* = 0.02) and *CD44-p53* (*p* = 0.01) co-occurred with positive co-expression of *CD44*, *iASPP*, and *p53*. In head and neck squamous cell carcinoma, invasive breast carcinoma, liver hepatocellular carcinoma, and kidney renal papillary cell carcinoma unaltered co-expression of *CD44*, *iASPP*, and *p53* favored tumor patient survival, whereas in glioblastoma multiforme, their altered expression was associated with tumor patient survival. We noticed that in the latter, *CD44* and *iASPP* were mutually exclusive, but *CD44* and *p53* co-occurred with positive co-expression of the three genes. Importantly, *CD44, iASPP*, and *p53* together did not play a significant role in survival of lung squamous cell carcinoma and adult soft tissue sarcoma (Appendix A).

### 3.7. CD44 Suppresses Cell Growth

As an adhesion molecule, CD44 promotes cell survival [41]. iASPP also promotes cell survival by inhibiting the apoptotic transactivation potential of p53 [27]. To explore the role of CD44 in the cell growth, we used non-immortalized human fibroblasts, AG1523, which respond better to external stimuli than hTERT-Bj fibroblasts. AG1523 cultures were depleted, or not, of CD44 and grown for several days until confluency. Cells in which CD44 had been knocked-down grew faster than CD44 expressing cells (Figure 6A) and reached a higher final cell density, with cells partially growing over each other (Figure 6B). Analysis by SDS-PAGE of the expression of p53 and the cyclin-dependent kinase inhibitor p27 [42] in AG1523 cells depleted of CD44 showed a decrease in p53 expression, mainly in high density (contact-inhibited) cultures, but an upregulation of p27 was seen mainly in sub-confluent (regular density) cultures (Figure 6C). Consistently, using ^3^H-thymidine-labelling of AG1523 cells, we noticed that depletion of CD44 resulted in increased cell proliferation which was counteracted by loss of iASPP in untreated or hyaluronan-treated cells (Figure 6D). Thus, CD44 suppresses the growth and survival of AG1523 cells.

Next, we investigated how external stimuli, such as Nutlin and FBS, affect CD44-iASPP complexes at regular and high-density cultures. As shown in Figure 6E, cell confluency did not affect the amounts of CD44-iASPP complexes in AG1523 cells; however, stimulation with 10% FBS enhanced the amounts of CD44-iASPP complexes. Furthermore, PDGF-BB significantly increased the interaction between CD44 and iASPP, but TGFβ stimulation did not have any appreciable effect on complex formation (Appendix A).

### 3.8. iASPP Is Required for Hyaluronan-Activated CD44-Induced Fibroblast Migration

To address the role of iASPP in hyaluronan-induced CD44-dependent cell migration, we studied the effect of silencing iASPP on migration of hTERT-BJ cells, treated or not, with hyaluronan or PDGF-BB. iASPP-silenced cells were unable to migrate in response to hyaluronan (Figure 7). Notably, PDGF-BB-mediated cell migration was only partly dependent on iASPP and was independent of CD44. Moreover, the addition of Hermes 1 antibodies, which block the binding of hyaluronan to CD44, suppressed hyaluronan-induced cell migration to control levels but had no effect on PDGF-BB-induced migration (Figure 7). Thus, iASPP is needed for hyaluronan-engaged CD44-induced, but not for PDGF-BB-induced, migration of hTERT-BJ fibroblasts.

### 3.9. p53 Is Required for CD44-Dependent Cell Adhesion

We then investigated the role of hyaluronan-engaged CD44 in AG1523 cells for the adhesion of cells to hyaluronan. As shown in Figure 8A, cells depleted of iASPP anchored to a hyaluronan substratum more strongly than cells expressing iASPP, suggesting that iASPP binding to the C-terminus of CD44 may decrease the binding of CD44 to ERM proteins and prevent actin polymerization and, thus, reduce cell adhesiveness. Treatment with the blocking antibody Hermes1 completely abolished cell adhesion, demonstrating a central role of hyaluronan binding to CD44 for fibroblast adhesion (Figure 8A). Importantly, p53-depleted cells exhibited a reduced adhesiveness independently of the external stimuli, suggesting a role of p53 in CD44-dependent fibroblast adhesion (Figure 8B).

### 3.10. CD44 Increases p53-Mediated ROS Production

Given that ROS inhibited the proliferation of normal cells in vitro through replicative senescence [43] and that the NADPH oxidase-generated ROS regulates the expression of CD44 [44], we investigated the role of CD44 and iASPP in the induction of ROS.

We transiently transfected CD44wt and iASPPwt-v5, individually and in combination, in HEK293 cells expressing high level of endogenous p53wt but negligible CD44s and moderate level of endogenous iASPP. We noticed that CD44wt, iASPPwt-v5, and endogenous p53wt formed complexes (Appendix A) and that the ectopic transfection of iASPPwt-v5 in the presence and absence of CD44wt reduced ROS production significantly. However, CD44wt expression elevated the p53-induced ROS level in the absence of iASPP in HEK293 cells (Figure 9).

## 4. Discussion

The present study reports a novel role of CD44 in regulating p53-mediated apoptosis and cell growth via interaction with the p53 inhibitor iASPP [45]. We found that the CD44s isoform bound iASPP and suppressed cell growth. Hyaluronan stimulation enhanced the formation of iASPP-CD44 complexes in fibroblasts but not in epithelial cells. Notably, we did not observe any complex between iASPP and CD44 variant (CD44v) isoforms. CD44v isoforms have been shown to limit ROS accumulation in glycolytic cancer cells with mutated p53, modulating glucose metabolism and promoting tumor growth [46,47]. This suggests that the different CD44 isoforms affect intracellular ROS levels and cell growth differentially.

Epithelial cells express high levels of CD44v isoforms [48]. Our finding that CD44v does not bind to iASPP suggests that hyaluronan might induce more iASPP-CD44 complexes in mesenchymal cells than in epithelial cells. Differential expression levels of iASPP and CD44 have been documented in different types of cancers, suggesting that iASPP-CD44 complexes occur at different levels in different types of cancers; further detailed analyzes are needed to elucidate the specific roles of iASPP-CD44 signaling in the progression of different cancers.

CD44 acts as a ‘picket’, connecting the cortical actin cytoskeleton to the plasma membrane via ezrin and ankyrin and, finally, with actin and spectrin, regulating clustering of other membrane receptors to elicit biological signaling [49]. Assembling CD44 into membrane–cytoskeletal junctional complexes is important in regulating cell adhesion, migration, and downstream signaling [10]. We observed significantly less interaction between iASPP and a CD44 mutant in which the ankyrin-binding domain had been mutated, whereas deletion of the C-terminal tail containing the PDZ domain and mutation of the known phosphorylation sites did not prevent iASPP-CD44 interaction.

Using proximity ligation assay, we investigated the cellular localization of endogenous CD44-iASPP complexes in situ; complexes were detected mainly in the cytoplasm. The fact that an interaction between CD44 and iASPP could be demonstrated in situ excludes the possibility that these complexes were artifacts formed during cell lysis.

We found that hyaluronan stimulation of CD44 enhanced the migration of fibroblasts in an iASPP-dependent manner. This finding is consistent with other recent studies that have demonstrated a role of iASPP in tumor progression by promoting migration [50]. Notably, iASPP can influence apoptosis irrespective of p53 status [51]. It is possible that iASPP exhibits different functions in different cell types.

Hyaluronan stimulation clusters CD44 molecules, but a self-association of CD44 also occurs in response to distinct signals, leading to activation of CD44. Such a self-association and, thus, activation of CD44 can also be promoted upon its ectopic overexpression in cultured cells. CD44 interacts with ERM proteins and regulates apoptosis through actin cytoskeletal rearrangements [22] and cooperates with NADPH oxidase and receptor tyrosine kinases to induce ROS [18,44]. Our findings that iASPP suppresses significant ROS activity (Figure 9) confirm previous observations that iASPP, independently of p53 expression, reduces ROS levels [52]. In this context, our finding that CD44 and iASPP suppress the generation of ROS in HEK293 cells suggests that these molecules may serve as therapeutic targets to suppress ROS generation in transformed cells where ROS production is high.

Our findings confirm previous observations that p53-dependent apoptotic signals depend on the repression of CD44, the expression of which otherwise regulates such signals [34]. The translocation of p53 and iASPP from the nucleus to the cytoplasm in CD44-silenced cells suggests an important regulatory role of CD44 since iASPP and p53 exhibit different functions in the nucleus compared with the cytoplasm.

## 5. Conclusions

We have reported that hyaluronan induces a complex between the standard isoform of CD44 and iASPP, preferentially in fibroblasts. Knock-down of CD44 resulted in an increased cell growth and cell density and to an increased amount of p53-iASPP complexes in the cytoplasm. We also found that iASPP is needed for hyaluronan/CD44-induced cell migration and adhesion. Our findings suggest that the balance between iASPP-CD44 and iASPP-p53 interactions affects cell migration and survival.

## Figures and Tables

**Figure 1 cancers-15-01082-f001:**
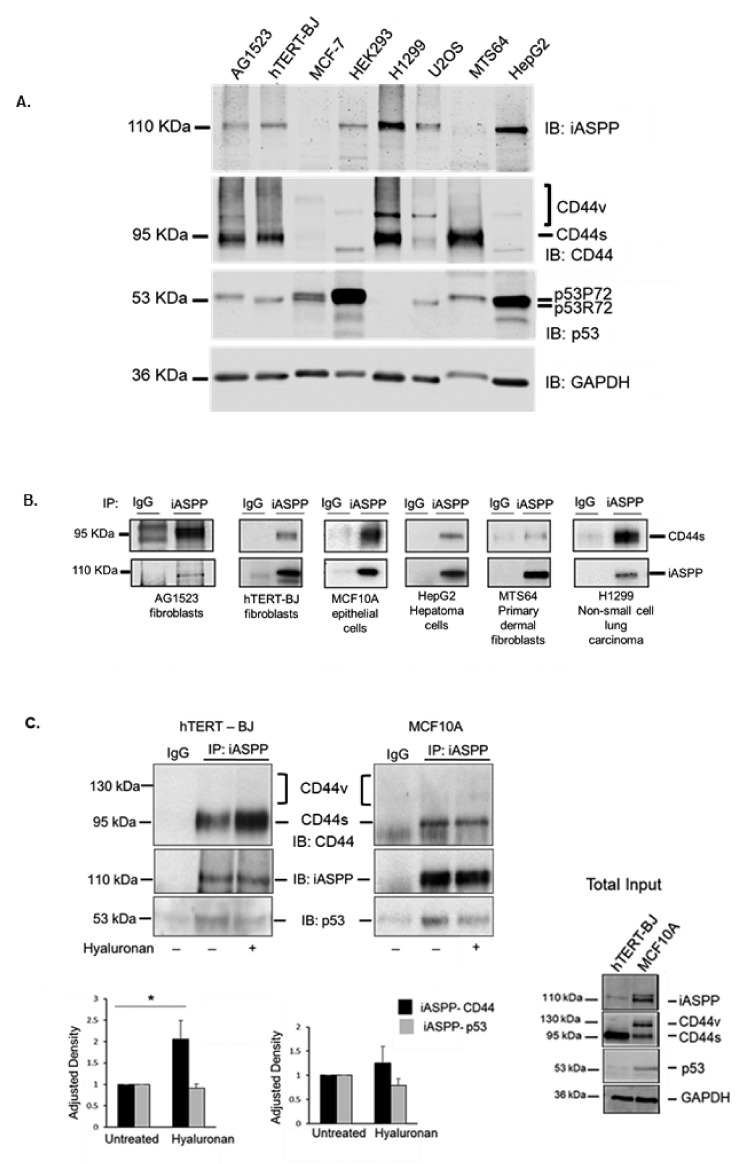
**CD44s forms physical complexes with iASPP, both in mesenchymal and epithelial cells.** (**A**) The expression levels of CD44, iASPP, and p53 were analysed in cell lysates from AG1523 fibroblasts, hTERT-BJ foreskin fibroblasts, breast cancer cells MCF-7, HEK293, lung carcinoma H1299, sarcoma U2OS, human dermal fibroblasts MTS64, and hepatoma HepG2 cells by immunoblotting (IB) with indicated antibodies, using anti-GAPDH as loading control. (**B**) Cell lysates from the selected cell lines were subjected to immunoprecipitation with an iASPP antibody or 3 µg/mL control rabbit IgG, followed by immunoblotting for CD44 and iASPP. (**C**) hTERT-BJ fibroblasts and MCF10A cells were cultured for 24 h in starvation medium prior stimulation, or not, with hyaluronan for 45 min. Cell lysates were then subjected to immunoprecipitation, using an iASPP antibody or control rabbit IgG, followed by immunoblotting with CD44, iASPP, and p53 antibodies. Note the absence of co-immunoprecipitated band at the expected position of CD44v just above the CD44s band. A representative experiment out of three performed is shown; lower panel in C indicates fold-change of mean values ± SD. of three independent experiments: * *p* < 0.05, Student’s *t*-test; significant difference compared with unstimulated cells. The uncropped blots are shown in Appendix A.

**Figure 2 cancers-15-01082-f002:**
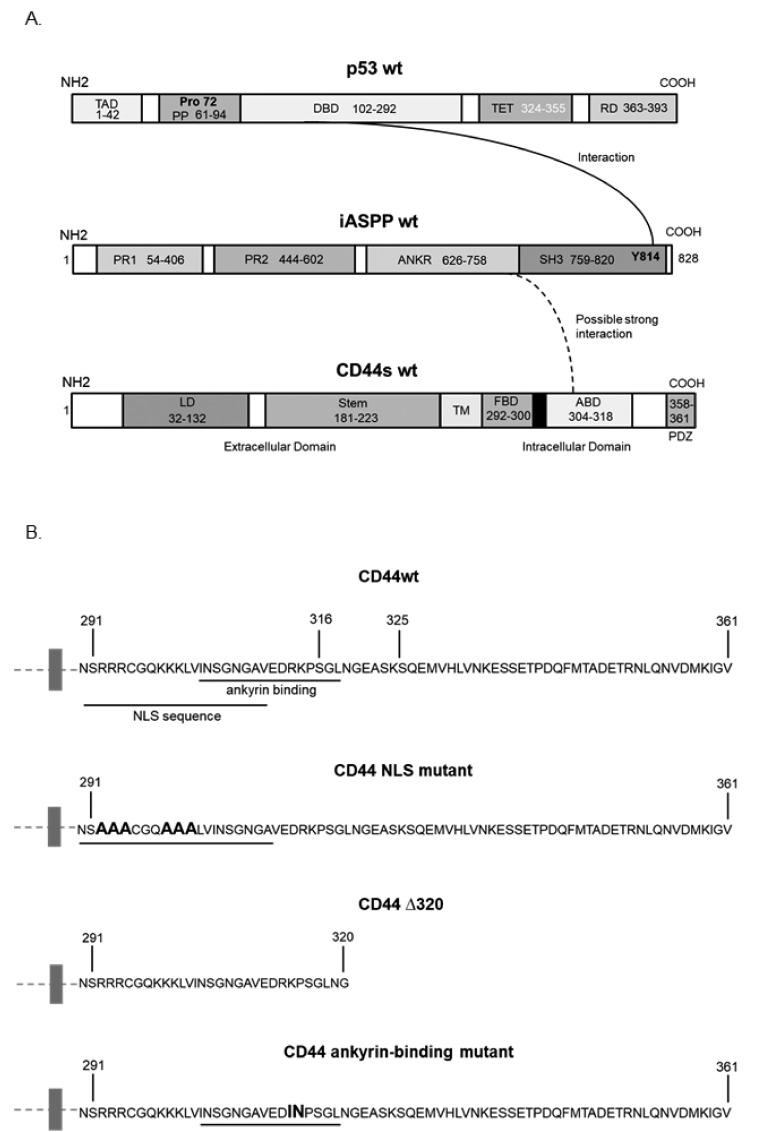
**Schematic illustration of the structures of p53, iASPP, and CD44, and CD44 mutants.** (**A**) An established interaction between domains in p53 and iASPP is indicated by a line, and the interaction found in the present study between iASPP and CD44s is indicated with a dashed line. (**B**) The cytosolic sequence of CD44 and the mutants used in this study are indicated.

**Figure 3 cancers-15-01082-f003:**
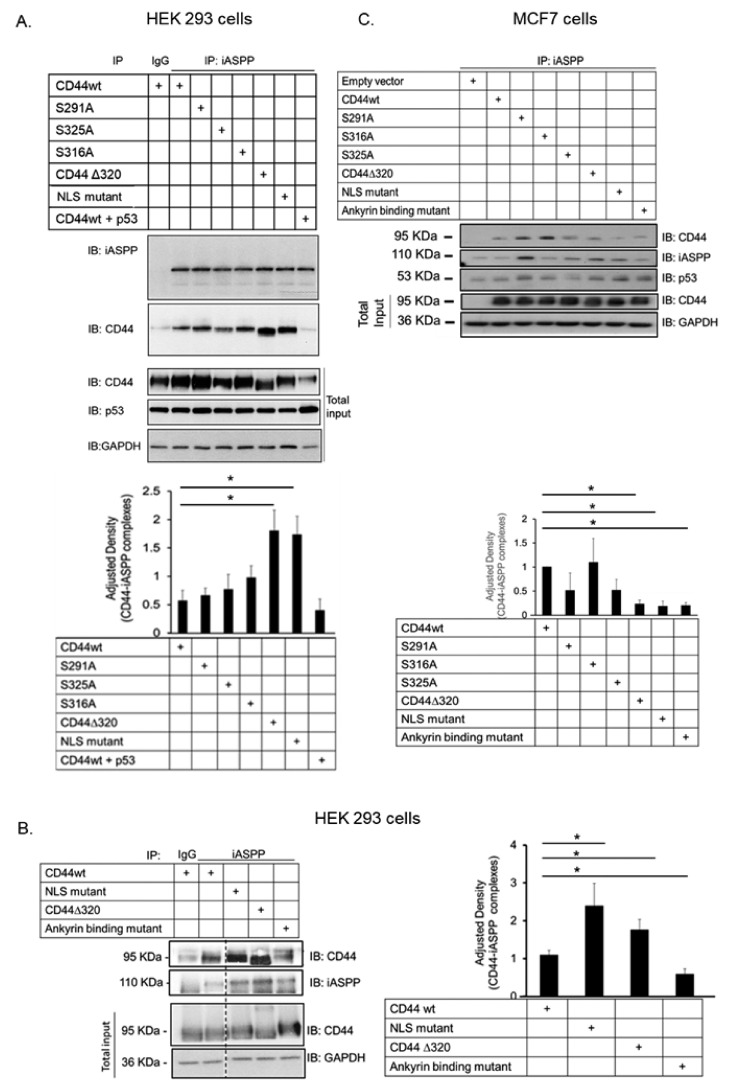
**CD44 ankyrin-binding domain interacts with iASPP.** (**A**,**C**). Wt CD44 and CD44s with mutated phosphorylation sites, ankyrin-binding domain and nuclear localization sequence, as well as a CD44 Δ320 deletion mutant, were transiently transfected in HEK293 cells (**A**,**B**) or MCF-7 cells (**C**). Cell lysates were subjected to immunoblotting using the indicated antibodies. Quantifications are shown below panels A and C, and to the right of panel B. The data are plotted in bar graphs representing the mean ± SD of at least three independent experiments (* *p* < 0.05, Student’s *t*-test). The uncropped blots are shown in Appendix A.

**Figure 4 cancers-15-01082-f004:**
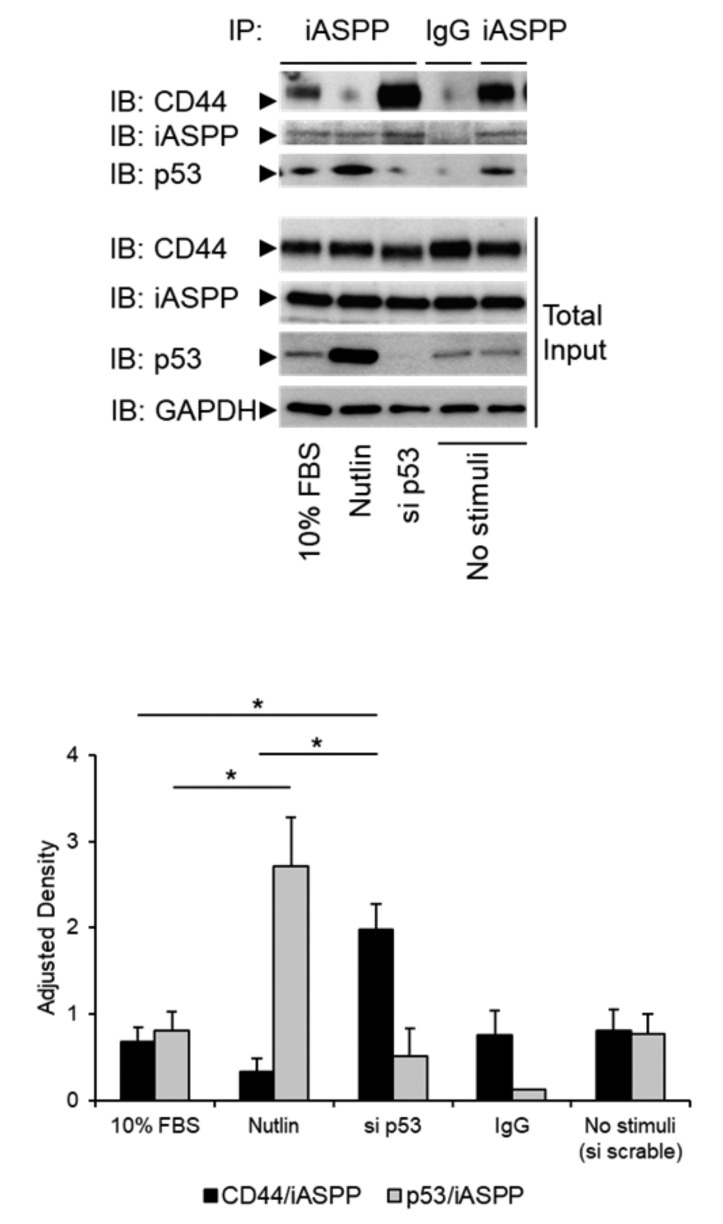
**p53 expression levels modulate formation of the complex between iASPP and CD44.** hTERT-BJ cells, treated with p53 siRNA, or not, were stimulated with 10% FBS or 10 nM Nutlin. Lysates were subjected to immunoprecipitation (IP) with an iASPP antiserum, followed by immunoblotting with antibodies against iASPP and p53. Quantification of the data is shown below the blot. The data are plotted in bar graphs representing the mean ± SD of at least three independent experiments (* *p* < 0.05, Student’s *t*-test). The uncropped blots are shown in Appendix A.

**Figure 5 cancers-15-01082-f005:**
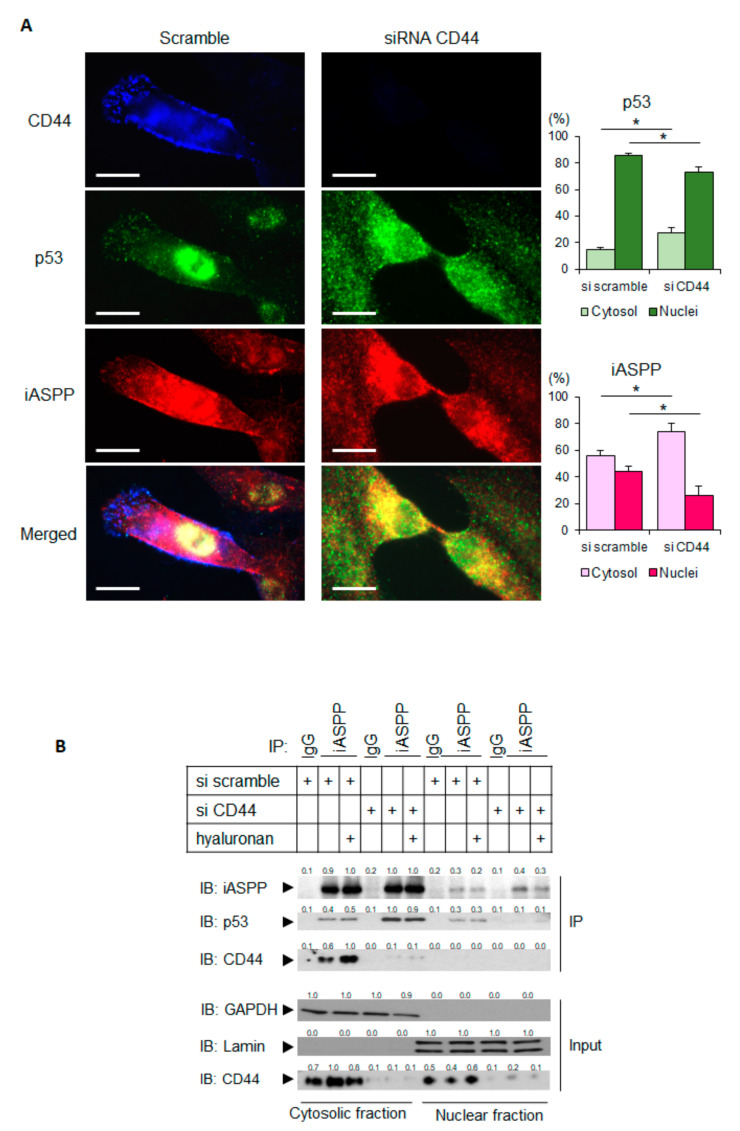
Depletion of CD44 promotes the translocation of p53 and iASPP to the cytoplasm. (**A**) hTERT-BJ cells treated with CD44 siRNA, or not, were stained for CD44 (blue), p53 (green), or iASPP (red). Images were taken using a Zeiss Axioplan 2 immunofluorescence microscope with a 63× objective. By ImageJ software, the localizations of p53 and iASPP in cytosol and nuclei were measured and are shown in the panels to the right. Scale bars, 10 µm. * *p* < 0.05, Student´s *t*-test. (**B**) Subcellular fractionation was performed on untreated or hyaluronan-stimulated hTERT-BJ cells transfected with siRNA for CD44 or scramble siRNA as control. Cell lysates were then immunoprecipitated with an iASPP antibody, followed by immunoblotting with iASPP, p53, and CD44 antibodies. The purity of the nuclear and cytoplasmic fractions was determined by immunoblotting for laminin and GAPDH, respectively. A representative experiment out of three independent experiments performed is shown. The data are plotted in bar graphs representing the mean ± SD of at least three independent experiments (* *p* < 0.05, Student’s *t*-test). The uncropped blots are shown in Appendix A.

**Figure 6 cancers-15-01082-f006:**
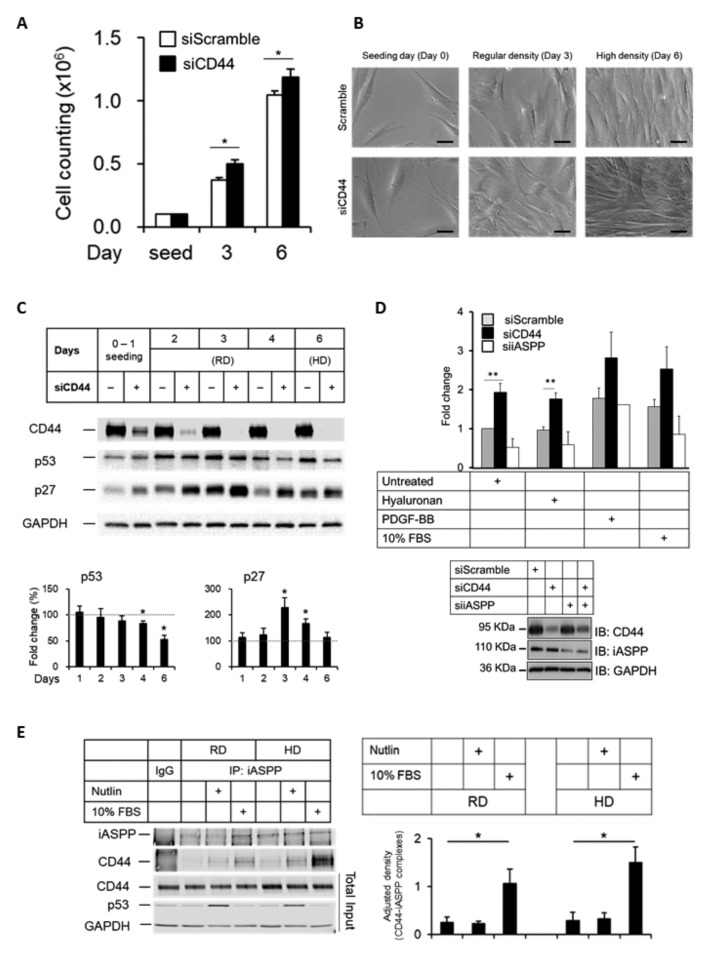
**Knock-down of CD44 in AG1523 fibroblast leads to loss of contact inhibition.** (**A**,**B**) AG1523 fibroblasts, transfected with CD44 siRNA or scramble siRNA, were seeded in 6-well plates and grown to a density of 10^5^ cells per well (regular density; RD) or >10^6^ cells per well (high density; HD), counted, and observed under microscopy. Scale bars, 20 µm. (**C**) Cell lysates from RD and HD cell cultures were subjected to immunoblotting using indicated antibodies. (**D**) AG1523 cells, treated with CD44 siRNA, iASPP siRNA, or scramble siRNA, were stimulated, or not, with hyaluronan, PDGF-BB, and 10% FBS, and then subjected to a ^3^H-thymidine incorporation assay. Knock-down efficiencies are shown in immunoblots. (**E**) Cell lysates from RD and HD cell cultures, incubated in Nutlin or 10% FBS, were subjected to immunoprecipitation using an iASPP antibody, followed by immunoblotting using indicated antibodies. The data are plotted in bar graphs representing the mean ± SD of at least three independent experiments (* *p* < 0.05 and ** *p* < 0.01, Student’s *t*-test). The uncropped blots are shown in Appendix A.

**Figure 7 cancers-15-01082-f007:**
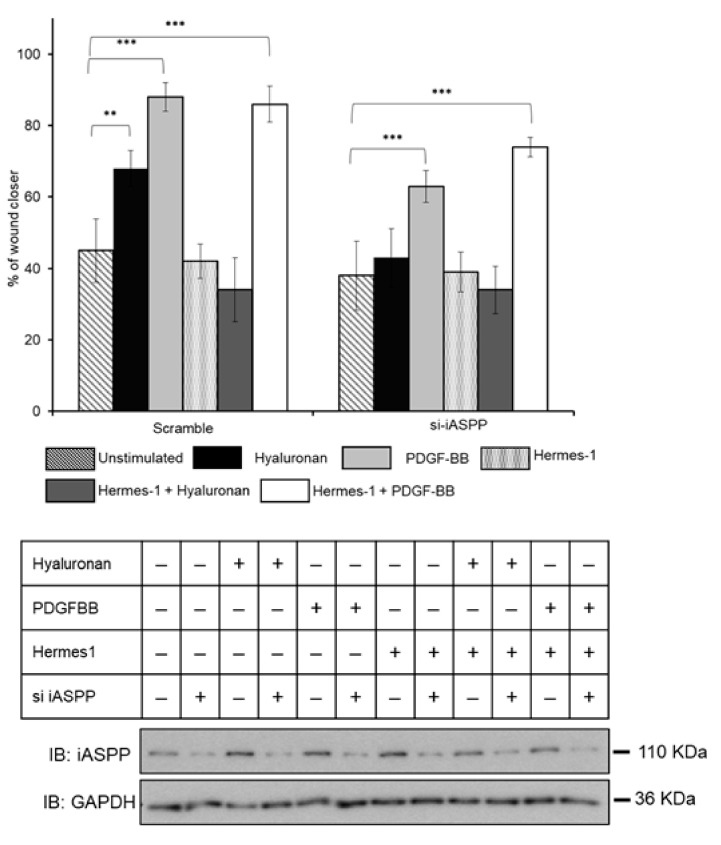
**iASPP is required for hyaluronan-induced fibroblast migration.** Confluent cultures of hTERT-BJ fibroblasts, transfected with iASPP siRNA or scrambled siRNA, were scratched and then stimulated, or not, with hyaluronan or PDGF-BB in DMEM medium containing 0.1% FBS. The wound size was examined after 24 h of culturing pictures were taken and analyzed using TScratch program (a software tool for automated analysis of wound-healing assays). The effect of hyaluronan stimulation and knock-down of iASPP on the migration of fibroblasts, are shown in bar graphs. The iASPP knock-down efficiency was determined by immunoblotting using an iASPP antibody; equal loading was determined by immunoblotting for GAPDH. The data are plotted in bar graphs representing the mean ± SD of at least three independent experiments (** *p* < 0.01 and *** *p* <0.001 Student’s *t*-test). The uncropped blots are shown in Appendix A.

**Figure 8 cancers-15-01082-f008:**
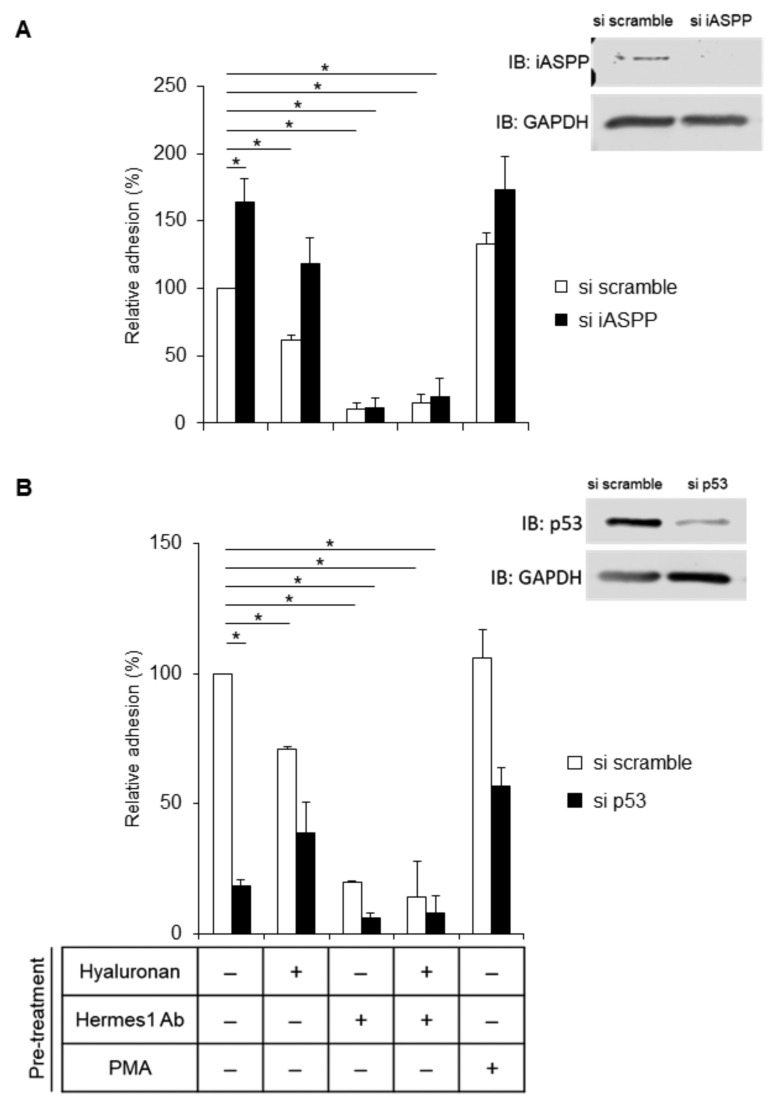
**iASPP suppresses but p53 increases CD44-mediated adhesion of fibroblasts.** (**A**,**B**) AG1523 fibroblasts incubated with iASPP siRNA (**A**), p53 siRNA (**B**), or scramble siRNA, were pre-treated, or not, with hyaluronan and blocking CD44 antibody or PMA and then seeded (30,000 cells per well) and incubated for 30 min. Adhered cells were then visualized by staining with crystal violet and quantified by measurement of the absorbance at 570 nm. The data are plotted in bar graphs representing the mean ± SD of at least three independent experiments (* *p* < 0.05, Student’s *t*-test). The uncropped blots are shown in Appendix A.

**Figure 9 cancers-15-01082-f009:**
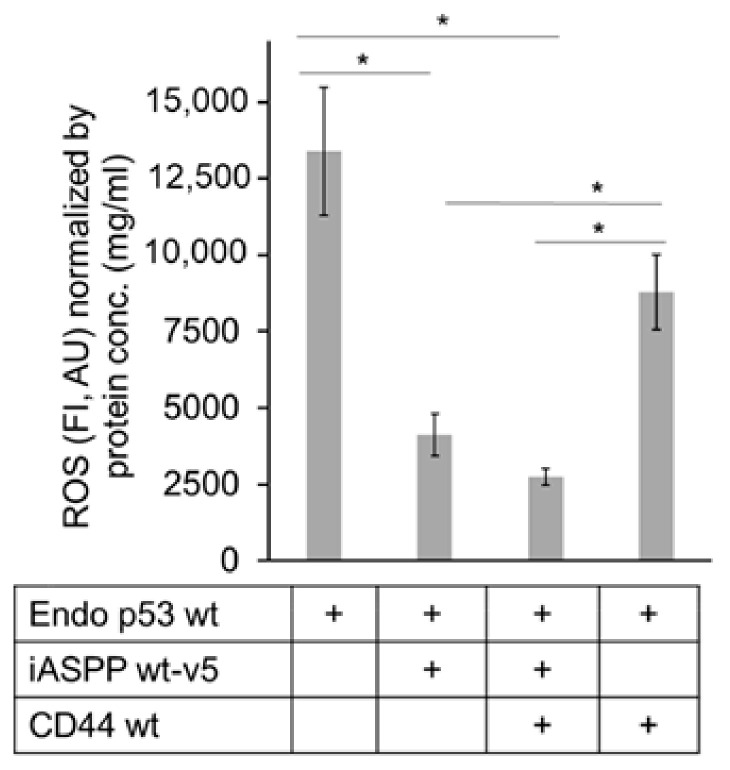
**iASPP-CD44 complexes suppress but CD44 increases p53-mediated ROS production.** V5-tagged wt iASPP (iASPP wt-v5) or CD44 were transfected individually or in combination in HEK293 cells expressing high level of endogenous p53wt, and the amount of ROS was determined. The data are plotted in bar graphs representing the mean ± SD of at least three independent experiments (* *p* < 0.05, Student’s *t*-test).

## Data Availability

The data presented in the study are available within the article and the Appendix A.

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
