# Peer review of "Hyaluronan-Induced CD44-iASPP Interaction Affects Fibroblast Migration and Survival"

_cancers, 2023, doi:10.3390/cancers15041082_

Round 1

Reviewer 1 Report

Lin et al. describe interesting results demonstrating the role of hyaluronan (HA) and its receptor CD44 in regulating fibroblast migration and survival. First, they determined that CD44s, not its variant forms, bind to iASPP, the p53 inhibitor. High molecular weight HA (HMW HA) enhanced this interaction in mesenchymal, but not epithelial, cells. The binding of iASPP to CD44 was via the ankyrin-binding domain, and overexpression of p53 decreased the association of iASPP and CD44. Further, CD44 was required for nuclear localization of p53-iASPP complexes. The associations of CD44, iASPP, and p53 with patient outcomes in various tumors were variable. CD44 expression was associated with suppressed growth and survival of at least one cell line without altering the interaction of CD44 and iASPP. PDGF-BB, but not TGFß, increased CD44-iASPP interaction. Interestingly, iASPP was necessary for HMW HA-stimulated but not PDGF-BB-stimulated migration, and blockade of CD44 binding to HA inhibited HA-stimulated but not PDGF-BB-stimulated migration. Expression of p53 increased CD44 binding to HA and was associated with increased ROS production, dependent on iASPP and CD44. The paper presents novel data on the contributions of CD44 to the regulation of migration and survival of fibroblasts. However, the following issues arise from the review of this manuscript.

1. While the detailed nature of the studies using multiple cell lines to define the interactions being studied is laudable, the manuscript reads like a series of observations rather than a flowing story. Perhaps creating a focused approach that demonstrates the yin-yang of iASPP with either CD44 or p53 as a critical regulatory step in controlling migration and survival would make the paper more readable.

 2.     Figure 5: The immunofluorescence images in A. do not match the quantitation in B. For example, the images show a dramatic decrease in nuclear staining, but the graphs show only a modest change in p53 localization.

 3.     Figure 7: No significance is indicated in the figure, whereas it is stated in the text.

 4.     Figure 9: Both iASPP and CD44 decrease ROS production compared to p53 WT. The title of the figure states CD44 increases p53-mediated ROS production.

 5.     Throughout the manuscript, it should be made clear that all data concerning HA is with HMW HA.

 6.     All the figures are small and difficult to read and interpret. These should be made larger and more legible.

 7.     Acknowledgements: This should either be deleted or the proper acknowledgments included.

Author Response

Reviewer #1

The manuscript titled "Hyaluronan-induced CD44-iASPP interaction affects fibroblast migration and survival" is a well-designed and well-executed study. However, the authors should consider the following suggestions for improving the work and its presentation:

  1. In the iASPP-CD44 interaction blot, please use a blot that spans over both CD44s and CD44v protein bands, it seems like the authors cut out the CD44s band alone. The supplementary blots also looks the same.

We thank the Reviewer for pointing this out. Now, we have replaced Fig. 1C and Fig. S1 with a new figure showing the position where the CD44v immunoreactive bands would be expected to be. In addition, the uncropped immunoblots are shown in Fig. S6.

  1. In the results section 3.2, the authors claim that the difference in the amount of CD44-iASPP complex formation in mesenchymal vs. epithelial cells depends on different signals. However, the authors did not test if this difference is because of the higher CD44 expression in mesenchymal cells as opposed to epithelial cells. The authors could test the hypothesis by ectopic overexpression in these cell types.

The Reviewer raises a valid point. We have reformulated the text to avoid over-interpretation (section 3.2, lines 300-301).

  1. In results section 3.4, line 333-337 is the elaboration of line 330-333 and talks about the same experiment. Avoid repetition and keep the appropriate explanation. When talking about the overexpression of any protein, establish how the overexpression is done to avoid mis-representation.

Fig. 3A-C shows results from two different cell types, i.e. HEK293 cells and MCF-7 cells. We feel that it is relevant to display data from more than one cell type. For clarity, we have now indicated the cell types used in the panels of Fig.3.

  1. In results section 3.5, I am not convinced of an exclusive co-localization of p53 and iASPP in the nucleus, the authors haven't provided any quantification of this co-localization. However, a significantly high co-localization is indeed there. Avoid over-interpretation of results.

We thank the Reviewer for this comment and we have now reformulated the sentence (line 377).

  1. In results section 3.6, Citing the figure/table at the first instance of dicussing the result will be helpful for the readers.

We have followed the Reviewer´s advice and cited the Figure and Table earlier.

  1. In the discussion, the authors claim, CD44s-iASPP complex reduced p53-induced ROS production and cell growth inhibition. However, the authors haven't addressed if iASPP expression alone is capable of inhibiting ROS production and growth inhibition, nor the authors proved the complexation is central to these phenotypes. It defenitly is a possible explanation but the authors could have explored the utility of the CD44s mutants to prove their claim. Especially, since there is an existing literature that suggest the ROS production inhibition by iASPP by Wenjie Ge etal, 2017 (https://doi.org/10.1016/j.ccell.2017.09.008) - also cite and discuss the work. 

We thank the reviewer for this comment and have revised the text to avoid over-interpretation (Section 3.10, line 495). We have also inserted a comment and a reference in the Discussion (lines 577-578).

  1. The authors could have utilized the CD44s mutants to establish the role of the complexes in cell phenotypes such as, cell growth, migration and ROS production. 

We agree with the Reviewer that such experiments could be of interest and we hope to be able to explore this further in the future. However, we feel that such an investigation is beyond the scope of the present study.

  1. Check for inconsistencies like iASSP in line 305.

We thank the Reviewer for having observed this mistake, which now has been corrected.

Reviewer 2 Report

The manuscript titled "Hyaluronan-induced CD44-iASPP interaction affects fibroblast migration and survival" is a well designed and well executed study. However, the authors should consider the following suggestions for improving the work and its presentation:

1. In the iASPP-CD44 interaction blot, please use a blot that spans over both CD44s and  CD44v protein bands, it seems like the authors cut out the CD44s band alone. The supplementary blots also looks the same.

2. In the resulst section 3.2, the authors claim that the difference in the amount of CD44-iASPP complex formation in mesenchymal vs. epithelial cells depends on different signals. However, the authors did not test if this difference is because of the higher CD44 expression in mesenchymal cells as opposed to epithelial cells. The authors could test the hypothesis by ectopic overexpression in these cell types.

3. In results section 3.4, line 333-337 is elaboration of line 330-333 and talks about the same experiment. Avoid repetition and keep the appropriate explanation. When talking about overexpression of any protein, establish how the overexpression is done to avoid mis-representation.

4. In results section 3.5, I am not convinced of an exclusive co-localization of p53 and iASPP in the nucleus, the authors haven't provided any quantification of this co-localization. However, a significantly high co-localization is indeed there. Avoid over-interpretation of results.

5. In results section 3.6, Citing the figure/table at the first instance of dicussing the result will be helpful for the readers.

6.  In the discussion, the authors claim, CD44s-iASPP complex reduced p53-induced ROS production and cell growth inhibition. However, the authors haven't addressed if iASPP expression alone is capable of inhibiting ROS production and growth inhibition, nor the authors proved the complexation is central to these phenotypes. It defenitly is a possible explanation but the authors could have explored the utility of the CD44s mutants to prove their claim. Especially, since there is an existing literature that suggest the ROS production inhibition by iASPP by Wenjie Ge etal, 2017 (https://doi.org/10.1016/j.ccell.2017.09.008) - also cite and discuss the work. 

7. The authors could have utilized the CD44s mutants to establish the role of the complexes in cell phenotypes such as, cell growth, migration and ROS production. 

8. Check for inconsistencies like iASSP in line 305.

Author Response

Reviewer #2

This manuscript investigates the mechanisms behind the hyaluronan induced complex formation of apoptosis-stimulating protein of p53 (iASPP) with the hyaluronan receptor CD44 in fibroblasts. This complex formation competes with the subcellular translocation of iASPP-p53 and leads to functional changes such as adhesion, migration, growth, and p53-mediated apoptosis. This is a well-designed and executed study, with great integration of various cell line studies and compelling observations.

Major comments

  1.  

Multiple cell lines were examined in this paper with compelling observations showing the functional consequences of iASPP-CD44. However, a singular cell line was used to analyze certain cell behavior changes. For example, cell migration assay and reactive oxygen species (ROS) measurements were performed in hTERT-BJ fibroblasts. The effects of knocking down CD44 on iASPP-p53 translocation and cell growth were examined in AG1523 fibroblasts. This may lead to the overgeneralization of mechanisms and the authors should expand upon it in the discussion.

The Reviewer raises an important point. In many of our experiments, we have used immortalized hTERT-BJ fibroblasts, which is convenient in a vitro cell model. However, for some experiments, e.g. the 3H-thymidine assays, we have used non-immortalized human fibroblasts, AG1523, which respond better to external stimuli. A comment addressing this point has been inserted in the text (section 3.7, lines 421-422).

Minor comments

  1. IM7 is the anti-CD44 antibody (line 268). It was introduced in supplementary results but not in the main article. Details should be included in the methods section. 

We thank the Reviewer for this observation. Information about IM7 antibodies has now been inserted in Table S1.

  1. In figure 1B, the caption on the IP for human dermal fibroblasts can include “MTS64” to be consistent in style with rest of the labels

We thank the Reviewer for this observation. Fig. 1B has now been changed.

  1. In figure 3, the label in 3A “NIS” should be “NLS”. 

This mistake has now been corrected.

  1. In figure 5A, the difference in nuclei vs cytosol expression for p53 seems significant. If so, the graph can be labeled similarly to iASPP to show the stats. 

The Reviewer is right. Fig. 5A has now been changed accordingly.

  1. The introduction of p27 in 397 was abrupt. Presenting some background literature prior will help. 

p27 has now been better introduced (line 426).

  1. Typical scratch wound assay is presented with images showing the gaps before and after the cell migrated, in addition to quantitative measurements. It will be helpful if the authors can add those images to supplemental figures.

We agree with the Reviewer that microscopic pictures of the scratch assay could have been shown. Unfortunately, this experiment was performed some time ago and the pictures were stored on a computer that is no longer operational. Thus, regrettably, we have only access to the quantification of these experiments. Fig 7, has been redrawn and a significance analysis has been inserted.

Reviewer 3 Report

This manuscript investigates the mechanisms behind hyaluronan induced complex formation of apoptosis-stimulating protein of p53 (iASPP) with the hyaluronan receptor CD44 in fibroblasts. This complex formation competes with the subcellular translocation of iASPP-p53 and leads to functional changes such as adhesion, migration, growth and p53-mediated apoptosis. This is a well designed and executed study, with great integration of various cell line studies and compelling observations.

Major comments

  1. Multiple cell lines were examined in this paper with compelling observations showing the functional consequences of iASPP-CD44. However, a singular cell line was used to analyze certain cell behavior changes. For example, cell migration assay and reactive oxygen species (ROS) measurements were performed in hTERT-BJ fibroblasts. The effects of knocking down CD44 on iASPP-p53 translocation and cell growth were examined in AG1523 fibroblasts. This may lead to overgeneralization of mechanisms and the authors should expand upon it in the discussion.

Minor comments

  1. IM7 is the anti-CD44 antibody (line 268). It was introduced in supplementary results but not in the main article. Details should be included in the methods section. 

  2. In figure 1B, the caption on the IP for human dermal fibroblasts can include “MTS64” to be consistent in style with rest of the labels

  3. In figure 3, the label in 3A “NIS” should be “NLS”. 

  4. In figure 5A, the difference of nuclei vs cytosol expression for p53 seems significant. If so, the graph can be labeled similar to iASPP to show the stats. 

  5. The introduction of p27 in 397 was abrupt. Presenting some background literature prior will help. 

  6. Typical scratch wound assay is presented with images showing the gaps before and after the cell migrated, in addition to quantitative measurements. It will be helpful if the authors can add those images to supplemental figures.

Author Response

Reviewer #2

This manuscript investigates the mechanisms behind hyaluronan induced complex formation of apoptosis-stimulating protein of p53 (iASPP) with the hyaluronan receptor CD44 in fibroblasts. This complex formation competes with the subcellular translocation of iASPP-p53 and leads to functional changes such as adhesion, migration, growth and p53-mediated apoptosis. This is a well designed and executed study, with great integration of various cell line studies and compelling observations.

Major comments

  1.  

Multiple cell lines were examined in this paper with compelling observations showing the functional consequences of iASPP-CD44. However, a singular cell line was used to analyze certain cell behavior changes. For example, cell migration assay and reactive oxygen species (ROS) measurements were performed in hTERT-BJ fibroblasts. The effects of knocking down CD44 on iASPP-p53 translocation and cell growth were examined in AG1523 fibroblasts. This may lead to overgeneralization of mechanisms and the authors should expand upon it in the discussion.

The Reviewer raises an important point. In many of our experiments, we have used immortalized hTERT-BJ fibroblasts, which is convenient in a vitro cell model. However, for some experiments, e.g. the 3H-thymidine assays, we have used non-immortalized human fibroblasts, AG1523, which respond better to external stimuli. A comment addressing this point has been inserted in the text (section 3.7, lines 421-422).

Minor comments

  1. IM7 is the anti-CD44 antibody (line 268). It was introduced in supplementary results but not in    the main article. Details should be included in the methods section. 

We thank the Reviewer for this observation. Information about IM7 antibodies has now been inserted in Table S1.

  1. In figure 1B, the caption on the IP for human dermal fibroblasts can include “MTS64” to be consistent in style with rest of the labels

We thank the Reviewer for this observation. Fig. 1B has now been changed.

  1. In figure 3, the label in 3A “NIS” should be “NLS”. 

This mistake has now been corrected.

  1. In figure 5A, the difference of nuclei vs cytosol expression for p53 seems significant. If so, the graph can be labeled similar to iASPP to show the stats. 

The Reviewer is right. Fig. 5A has now been changed accordingly.

  1. The introduction of p27 in 397 was abrupt. Presenting some background literature prior will help. 

p27 has now been better introduced (line 426).

  1. Typical scratch wound assay is presented with images showing the gaps before and after the cell migrated, in addition to quantitative measurements. It will be helpful if the authors can add those images to supplemental figures.

We agree with the Reviewer that microscopic pictures of the scratch assay could have been shown. Unfortunately, this experiment was performed some time ago and the pictures were stored on a computer that is no longer operational. Thus, regrettably, we have only access to the quantification of these experiments. Fig 7, has been redrawn and a significance analysis has been inserted.
